# Enhancing Thickness Uniformity of Nb_2_O_5_/SiO_2_ Multilayers Using Shadow Masks for Flexible Color-Filtering Applications

**DOI:** 10.3390/mi15040551

**Published:** 2024-04-21

**Authors:** Tzu-Chien Li, Dong-Lin Li, Jiashow Ho, Chih-Chiang Yu, Sheng-Shih Wang, Jyh-Jier Ho

**Affiliations:** 1Department of Electrical Engineering, National Taiwan Ocean University, No. 2, Peining Rd., Keelung 20224, Taiwan; nike716107@gmail.com (T.-C.L.); ericli@email.ntou.edu.tw (D.-L.L.); sswang0410@gmail.com (S.-S.W.); 2Department of Electrical Engineering, University of California, 66-147B Eng. IV Building, Los Angeles, CA 90095-1594, USA; jth72507@gmail.com

**Keywords:** shadow mask, ion-assisted deposition (IAD), color filter (CF), uniform thickness, flexible substrate, 50% transmittance (T50%)

## Abstract

Using a stainless shadow mask combined with a magnetron-ion-assisted deposition (IAD) sputtering system, we investigate the surface morphologies and optical properties of microfilms. Optimal color-filter (CF) coating microfilms with niobium pent-oxide (Nb_2_O_5_)/silicon dioxide (SiO_2_) multilayers on a hard polycarbonate (HPC) substrate, grown at 85 °C and 50 SCCM oxygen flow, can obtain a fairly uniform thickness (with an average roughness of 0.083 and 0.106 nm respectively for Nb_2_O_5_ and SiO_2_ films) through all positions. On a flexible HPC substrate with the Nb_2_O_5_/SiO_2_ microfilms, meanwhile, the peak transmittances measured in the visible range are 95.70% and 91.47%, respectively, for coatings with and without a shadow mask for this new-tech system. For the optimal CF application with a shadow mask, transmittance on each 100 nm band-pass wavelength is enhanced by 4.04% absolute (blue), 2.96% absolute (green), and 2.12% absolute (red). Moreover, the developed new-tech system not only enhances the quality of the films by achieving smoother and uniform surfaces but also reduces deposition time, thereby improving overall process efficiency. For the with-shadow-mask condition, there is little shift at 50% transmittance (T50%), and high transmittance (~97%) is maintained after high-temperature (200 °C) baking for 12 h. These results are well above the commercial CF standard (larger than 90%) and demonstrate reliability and good durability for flexible optical applications.

## 1. Introduction

In the realm of sputtering technologies for depositing microfilms, achieving uniformity in the surface morphologies is paramount for ensuring optimal performance across various applications [1,2,3,4,5]. Two problems generally occur in terms of the uniform distribution of the thickness deposited by the direct current (dc) reactive magnetron sputtering system. The first problem consists of an electrically insulating layer generated on the target surface during the high-temperature sputtering process, resulting in arcing due to charge accumulation. In particular, insulating film deposition on the electrode strongly affects the gas discharge properties, not only by blocking the current flow but also by discharging the dielectric surface fully by reverse pulses [6]. Furthermore, a high-temperature process with low discharging limits flexible application on hard polycarbonate (HPC) or stainless-steel substrates.

In the second non-uniform problem from magnetron sputtering, the distance between the target and substrate is often very short, and as a result, the coating thickness at different positions of the substrate depends on the sputtering distribution of the target material [7,8]. Without controlling the uniformity, a coating on a flat substrate is usually thick at the center, and the thickness gradually decreases toward the edges [8,9]. This limits the size and number of coating elements for multi-film applications that can be produced in a single coating run. In the case of curved optics, the desired thickness distribution cannot be achieved without effectively controlling the thickness. Thus, controlling the lateral thickness distribution is essential for coatings prepared by magnetron sputtering.

To solve these problems, ion-assistance deposition (IAD) equipped with a shadow mask with an adjustable stripe and length can be attached to a reactive magnetic sputter [10,11,12,13], allowing deposition for optical applications with smooth-surface and high-density microfilms under low-temperature operation. By evaluating the different deposition thicknesses at individual positions on the film surface, the stripe length of the shadow mask is adjusted, thus improving the uniformity and smoothness of the film surface [14].

Our aims are to demonstrate the environmental durability of this new technique and to deposit a multilayer to verify the new-tech advantages in enhancing the performance of flexible color-filter (CF) applications. In this research, therefore, we not only investigate the technique of IAD combined with adjustable shadow masks to improve the surface uniformity of microfilms but also conduct durability evaluations on the developed flexible CFs. These harsh environmental conditions include immersion in boiling water and saltwater as well as exposure to high temperature to enhance the CF performance and surface-morphology improvement from new-tech application opportunities.

## 2. Experiments and Measurements

The block diagram of a reactive magnetron sputter equipped with an IAD system (as shown in Figure 1a) and the detailed thin stripe-length stainless-steel shadow mask (with 200 μm thickness, placed as shown in Figure 1b,c) to produce beneficial modifications in the surface morphologies for optical applications are provided [15]. In the developed IAD sputter in Figure 1a, there were niobium (Nb) and silicon (Si) targets, both with cylinder shapes, where cooling water flowed at the center during the sputtering process. A drum (at 60 rpm speed) under the 100 V pulse-dc voltage of discharging ions rotated each HPC substrate (with 30 × 30 mm^2^ dimension). The reactive sputtering also involved allowing sufficient reactive gases (Ar and O_2_) into the chamber to keep the target completely covered with a dielectric [16], for uniform film deposition of the Nb and Si materials with sequential and continuous coating.

This study uses a vacuum coating system equipped with an electron beam gun. At the ambient temperature of the substrate, niobium pent-oxide (Nb_2_O_5_) and silicon dioxide (SiO_2_) thin films are deposited on the substrate through an electron beam supported by oxygen ions (O^−^). Meanwhile, as shown in Figure 1b, we added the detailed stripe-length shadow mask at 16 positions for measurement, hoping to enhance the uniformity of the new-tech color-filtering microfilms. Furthermore, Figure 1c illustrates the sputtering geometry in which the substrate is mounted on a rotating drum; *β* is the emission angle of the ejected particle flux relative to the normal to the target surface, and *α* is the incidence angle of the deposited particle flux relative to the normal to the substrate surface. On path A, the particles reach the substrate, and on path B, the shadow mask blocks the particles. In order to optimize the mask to achieve high uniformity over a large area, the sputtering yield, the angular distribution of the ejected particles from the target (path A), the mask restriction function, the arriving angle (*α*) of the sputtered particles on the substrate, and the substrate movement need to be taken into account.

Therefore, the chamber was first pumped down to 1 × 10^−5^ Torr or less. Then, the working Ar (99.99%) gas was applied with a flow rate of 55 and 85 SCCM (1 SCCM is identical to 1 atm cm^3^/min at STP), respectively, for Nb and Si targets. Pure O_2_ gas (99.99%) with a 50 SCCM flow rate, 125 W sputtering power, and 85 °C deposition temperature could also be introduced into the vacuum system to assist oxidation during the deposition process (deposition rate was 2.04 and 3.43 nm/min, respectively, for Nb_2_O_5_ and SiO_2_ microfilms). The total working pressure was ~5.1 × 10^−4^ Torr, with mean 0.2 A sputtering current. The s-polarized component of the light transmitted by the coated substrate was monitored at 550 nm during deposition of the layers and at 5 nm intervals over the 400~700 nm spectral range after deposition.

To measure the optical properties of the developed CFs, we used a spectrophotometer (Hitachi UV-Visible-NIR, Tokyo, Japan) to measure the transmittance within the visible wavelength range (400–700 nm). Additionally, we employed a scanning electron microscope (SEM, Hitachi S-4100) and atomic force microscopic (AFM, Digital Instruments Inc. (Tonawanda, NY, USA), NanoScope E with dimension 3100 controller) system to observe the surface morphology in this experiment. The surface roughness (RMS for *R*q) values were obtained with the software that came with the instrument.

## 3. Results and Discussion

Table 1 lists the morphology comparison of Nb_2_O_5_ (at top, ~400 nm thickness) and SiO_2_ (at bottom, ~600 nm thickness) microfilms as deposited with (at right)/without (at left) shadow-mask conditions. The corresponding surface morphology (2D and 3D AFM images below), in which *R*max, *R*a, and *R*q stand for the maximum surface height, average centerline, and root-mean-square (RMS) roughness, respectively.

For the without-shadow-mask condition, Nb_2_O_5_ or SiO_2_ film images show a relatively compact and flat surface structure; the film was composed of cone-shaped columns randomly distributed over the surface of the film. Thus, the films’ surface was rough and non-uniform, with the maximum surface height (Rmax)/average centerline (Ra)/RMS (Rq) surface roughness of 8.778/0.760/0.974 nm and 5.360/0.488/0.631 nm, respectively, for Nb_2_O_5_ and SiO_2_ microfilms. For the with-shadow-mask condition, the Rmax/Ra/Rq values are 8.710/0.603/0.792 nm and 2.211/0.186/0.242 nm, respectively, for Nb_2_O_5_ and SiO_2_ microfilms, being much smoother and uniformly distributed over the film surface. Meanwhile, this shadow-mask design could adjust the stripe length in accordance with different positions, thus shortening the morphology roughness and improving surface uniformity. Shadow-mask sizes are typically adjusted for uniform optimization via the preferential sputtering caused by ion bombardment [16].

Figure 2 illustrates a comparison of the deposition thicknesses at various positions for Nb_2_O_5_ and SiO_2_ microfilms prepared with/without a shadow mask in the developed magnetron-IAD sputtering system. For comparison with/without a shadow mask, Nb_2_O_5_ microfilm-roughness ranges are large, from 19.86 to only 1.33 nm. Similarly, SiO_2_ microfilm-roughness ranges are large, from 34.5 to only 1.7 nm. For the condition with a stripe-length shadow mask, significantly, all the films’ surface morphologies were smooth and uniform, with less roughness, with a decrease in morphology roughness similar to the phenomena in the AFM images (values of *R*max, *R*a, and *R*q) of Table 1.

The roughness of SiO_2_ films is significantly less than that of Nb_2_O_5_ films. The surface morphology becomes rougher as the layer thickness increases, both with/without shadow mask conditions, as shown in the bottom-left axis of Figure 2. In Figure 2b, meanwhile, the SiO_2_ film thickness with the mask is higher in positions 13–16 than without the mask, which is consistent with the stripe-length trend (blue lines). This phenomenon may be attributed to the deposited material accumulating on the evolving edges of the micro-crystallites [18]. It can also illustrate that the adjustment trend of the stripe length is related to the thickness of the deposited film. Without masking, locations corresponding to thicker film deposition have shorter stripe length, while locations corresponding to thinner film deposition have longer stripe length.

For the without-shadow-mask condition (red lines in the bottom-left axis of Figure 2), it can be found that the thickness of the microfilm first increases corresponding to the position number from 1 to 8 and then decreases from the 9th to the 16th position. The microfilms’ surfaces were rough and non-uniform, with an average roughness of 1.241 and 2.156 nm, respectively, for Nb_2_O_5_ and SiO_2_ films. It can be inferred that the adjustment trend of the stripe length of the mask is related to the thickness of the microfilm deposition [9,14]. On the bottom-relation axes of Figure 2, therefore, the stripe length (**blue** line referred as the right-Y axis) at all positions should be adjusted by following this trend (**red** line referred as the left-Y axis). For the with-shadow-mask condition (**green** lines in the bottom-left axis of Figure 2), consequently, it can be verified by the above trend that a fairly uniform thickness is obtained (with an average roughness of 0.083 and 0.106 nm, respectively, for Nb_2_O_5_ and SiO_2_ films) through all corresponding positions.

The major CF design is a combination of high and low refractive indices, i.e., Nb_2_O_5_/SiO_2_ microfilms, of multilayer coatings [19]. The refractive indices (*n*) and extinction coefficient (*k*) can be calculated by Equations (1) and (2) [20,21]:(1)$$n={\left[Q+{\left(Q^{2}-n_{s}^{2}\right)}^{1/2}\right]}^{1/2}$$
and
(2)$$k=\left(\frac{\lambda }{4\pi d}\right)\left(\ln(x)\right)$$
where
$$\{\begin{array}{c} \begin{array}{l} Q=2\left(n_{s}\right)\left[\frac{T_{M}-T_{m}}{T_{M}T_{m}}\right]+\left(\frac{n_{s}^{2}+1}{2}\right) \end{array}\\ x=\frac{F-{\left[F^{2}-{\left(n^{2}-1\right)}^{3}\left(n^{2}-n_{s}^{2}\right)\right]}^{1/2}}{{\left(n-1\right)}^{3}\left(n-n_{s}^{2}\right)}\\ \begin{array}{l} F=\frac{8\left(n^{2}\right)n_{s}}{T_{i}},\;T_{i}=\frac{2\left(T_{M}\right)T_{m}}{T_{M}+T_{m}}\\ d=\frac{\left(\lambda_{1}\right)\left(\lambda_{2}\right)}{2\left(n_{2}\left(\lambda_{1}\right)-n_{1}\left(\lambda_{2}\right)\right)} \end{array} \end{array}$$
where *n*_s_ is the substrate refractive index; *T*_M_ and *T*_m_ are the maximum and minimum transmittance of the spectrum, respectively; *d* is the film thickness; *λ*_1_ and *λ*_2_ are the adjacent maximum and minimum wavelength, respectively; and *n*_1_ and *n*_2_ are the refractive indices corresponding to *λ*_1_ and *λ*_2_.

In the manufacturing of color-filtering applications, the layering process causes particles to settle between and on top of each layer. Consequently, all layers, including the CF coatings, have rough surfaces, posing challenges for accurately determining the (*n*, *k*, *d*) parameters through inverse methods due to scattering effects [19]. To obtain the (*n*, *k*, *d*) effective values simultaneously [19], mean square errors (MSEs) between measured and fitted reflectivity for a wide range of illuminations are minimized.

For the (*n*, *k*, *d*) effective values calculated by Equations (1) and (2), the optimal Nb_2_O_5_/SiO_2_-bilayer thickness can be determined for the layer numbers required to enhance the CF performance. Thus, Figure 3 and Figure 4 demonstrate the (*n*, *k*, *d*) relationship under the discharge current range (3–8 A) of the ion source and at *λ* = 450 nm with/without shadow masks, respectively, from the (*k*, *d*) and (*n*, *d*) parameter models. In the (*k*, *d*) model depicted in Figure 3, the roughness of the film thickness (*d*, in nm at the bottom-left axis) and the extinction coefficient (*k*, in ×10^−5^ order at the bottom-right axis) as a function at the various 16 positions, respectively, are given for (a) Nb_2_O_5_ and (b) SiO_2_ microfilms as compared with/without shadow masks. For the comparison with/without a shadow mask, the *k* values (blue lines) were increased by 0.019% (from 3.01 to 4.90 × 10^−4^ for Nb_2_O_5_ film) and 3.0 × 10^−3^% (from 1.42 to 1.45 × 10^−4^ for SiO_2_ film), respectively. In contrast, the *d* values (green lines) were, respectively, decreased by 18.53 nm (from 19.86 to only 1.33 nm for Nb_2_O_5_ film) and 32.8 nm (from 34.5 to only 1.7 nm for SiO_2_ film) as compared with/without a shadow mask in the developed IAD-sputter system. Therefore, the curve rate of change (in %) of k and d values is 0.7% or 0.8%, respectively, becoming smoother, with a uniform surface. Meanwhile, in Figure 3a,b, it can be seen that the k values of Nb_2_O_5_ films are higher than those for the films of SiO_2_. This is because SiO_2_ usually shows lower absorption characteristics in the visible spectrum range (400–700 nm wavelength), while Nb_2_O_5_ films can show higher absorption; the same argument can be found in [22].

Similarly, for the (*n*, *d*) parameters plotted in Figure 4, the difference of the film thickness (*d*, in nm at the bottom-left axis) and the refractive indices (*n*, at the bottom-right axis) as a function at the various 16 positions, respectively, are given for (a) Nb_2_O_5_ and (b) SiO_2_ microfilms as compared with/without shadow masks. It can be seen that the curve tendency of the *n* and *d* values becomes smoother, with a uniform surface, after using the shadow mask in the developed magnetron-IAD sputtering system. For the comparison with/without a shadow mask, the *n* values (blue lines) were increased by 0.045% (from 2.36419 to 2.36464 for Nb_2_O_5_ film) and 3.98% (from 1.5055 to 1.5453 for SiO_2_ film), respectively.

It can be observed that the *n* value of the film without the shadow mask increases as the *d* values increase, showing a proportional relationship consistent with the literature [23]. In contrast, since the *d* values (green lines) of the films produced with mask remain uniform, the change in *n* values is quite small, with only 0.011% variation (from 2.36390 to 2.36410 for Nb_2_O_5_ film) and 3.98% variation (from 1.5055 to 1.5453 for Si_2_O_2_ film), respectively, as compared with/without a shadow mask in the developed IAD-sputter system.

Meanwhile, in Figure 4a,b, it can also be seen that the *n* values of Nb_2_O_5_ films are higher than those for the films of SiO_2_. This is because SiO_2_ is a non-metal oxide with a simpler structure, and its refractive index is usually lower. In contrast, Nb_2_O_5_ is a metal oxide with a complex structure composed of niobium and oxygen atoms, and its refractive index may be higher. This is the same argument as found in [24].

Following this (*n*, *k*, *d*) effective model, we can evaluate the optimal Nb_2_O_5_/SiO_2_-bilayer thickness and determine the layer numbers required to enhance the CF performance. Thus, Figure 5a–c depict the layer-number evaluation (19~37 layers) required for the narrow band-pass wavelength, respectively, for the blue (B), green (G), and red (R) regions. At the same time, in Figure 5, the optimal number of 36 layers can be determined from the narrow bandwidth (100 nm wavelength) for (a) B region (400–500 nm wavelength), (b) G region (500–600 nm wavelength) and (c) R region (600–700 nm wavelength).

The values of the optimal 36-multilayer (Nb_2_O_5_/SiO_2_-bilayer) thicknesses of the RGB CF coatings determined from the effective (*n*, *k*, *d*) model of Figure 5a–c are listed in Table 2a. The total 36-multilayer thicknesses are 2.27, 2.86, and 3.11 μm for the RGB CF coatings. Table 2b plots the optimal 36-layer CFs using a flexible HPC substrate (30 × 30 mm^2^ dimension) and the developed magnetron-IAD with a shadow-mask sputtering system.

To verify the shadow-mask effect on the surface uniformity of microfilms, we compare the 36-layer transmittance for all RGB CFs sputtered by the developed system. For each color filter using the shadow mask, Table 3 shows that there was a great enhancement in the transmittance on each 100 nm band-pass wavelength by 4.04%absolute (blue CF with 3.11 μm thickness), 2.96%absolute (green CF with 2.86 μm thickness), and 2.12% absolute (red CF with 2.21 μm thickness). This shadow-mask CF design can also improve the narrow-band transmitting property, which corresponds to enhancing the surface uniformity, as presented in the AFM images of Table 1. Thus, it is the surface-roughness effect of the CF microfilm at a given wavelength, the same argument as found in [25], that is related to its high transmittance.

In the process of CF development, conducting stability verification before mass production is crucial. This phase ensures that the product can withstand various harsh conditions and meets the expected performance standards. Evaluation and testing in hot water, saltwater, and high temperatures were conducted on the flexible CFs with a Nb_2_O_5_/SiO_2_ multilayer coating.

Figure 6 illustrates the changes in light transmittance of the CFs after high temperature (200 °C) from 0 to 12 h. In Figure 6a, it can be observed that, in the meantime, the T50% (at 50% transmittance) relative transmittance wavelength of the filters prepared as unmasked shifts from 662 to 638 nm wavelength (decreasing by 3.63%), and the transmittance decreases from 97% to 88% (variation of 9%absolute). In comparison, the CF in Figure 6b maintains the transmittance and wavelength at 662 nm (T50% variation for 0.3%absolute) and 97%, respectively.

After immersion in boiling (100 °C) water for 4 h and salt (100%) water for 3 days, the developed CFs with the shadow mask showed variations at T50% of 4.39 and 2.16%absolute, respectively, which can be considered negligible (both less than 5%). Therefore, all the phenomena demonstrate that CFs deposited with the shadow mask are more stable under harsh environments for optical applications. This also demonstrates that this new-tech system can enhance CF durability and reliability.

## 4. Conclusions

In summary, the adjustable stripe lengths of the shadow mask with the magnetron-IAD sputtering system can enhance the surface thickness uniformity of Nb_2_O_5_/SiO_2_ microfilms. The important novelty of this study is that we constructed a (*n*, *k*, *d*) effective model and calculated the simulated reflectance spectra by incorporating the variable refractive index and thickness of each layer of Nb_2_O_5_/SiO_2_ microfilms. Following the (*n*, *k*, *d*) parameter model, the optimal Nb_2_O_5_/SiO_2_-bilayer thickness can be evaluated, and layer numbers required to enhance the CF performance can be determined.

The optimal 36-layer CF was fabricated and observed by AFM and SEM to determine surface morphology. Next, the transmittance was measured by a spectrophotometer. Thus, for each CF with the shadow mask, each 100 nm band-pass wavelength exhibited a great enhancement in the transmittance, by 4.04%absolute (blue CF), 2.96%absolute (green CF), and 2.12% absolute (red CF). Finally, the developed new-tech system was shown to improve both the performance and surface morphology of the flexible-CF application, thus ensuring great durability under harsh environmental conditions (such as hot-temperature baking, saltwater immersion, and boiling water treatment). All these results are well above the commercial CF standard (larger than 90%), showing the reliability and durability of this developed new-tech system.

## Figures and Tables

**Figure 1 micromachines-15-00551-f001:**
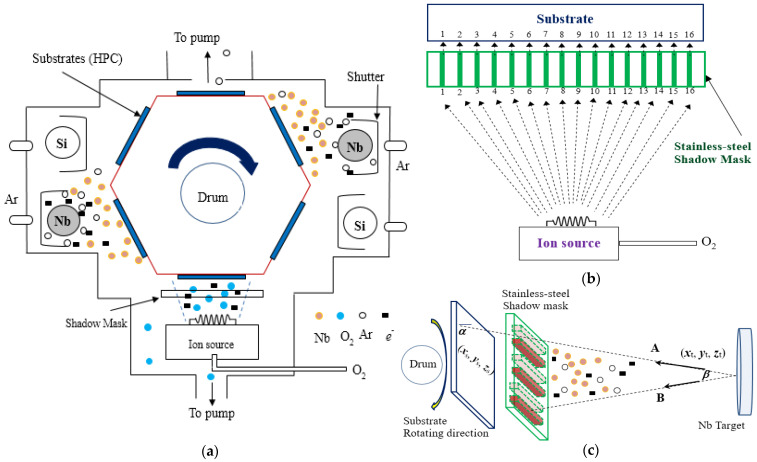
(**a**) Schematic diagram of the developed reactive magnetron sputter equipped with the IAD system and a shadow mask. (**b**) The detailed adjustable 16 positions of thin stripe-length stainless-steel shadow mask. A numerical optimal algorithm with trial–error iteration [17] optimizes the design of shadow-mask size (with 30 × 30 mm^2^ dimension and 200 μm thickness). (**c**) The orientation of the shadow mask relative to the targets. This takes into account the substrates on the rotating drum, sputtering targets, and a shadow mask with predefined stripe-length shape.

**Figure 2 micromachines-15-00551-f002:**
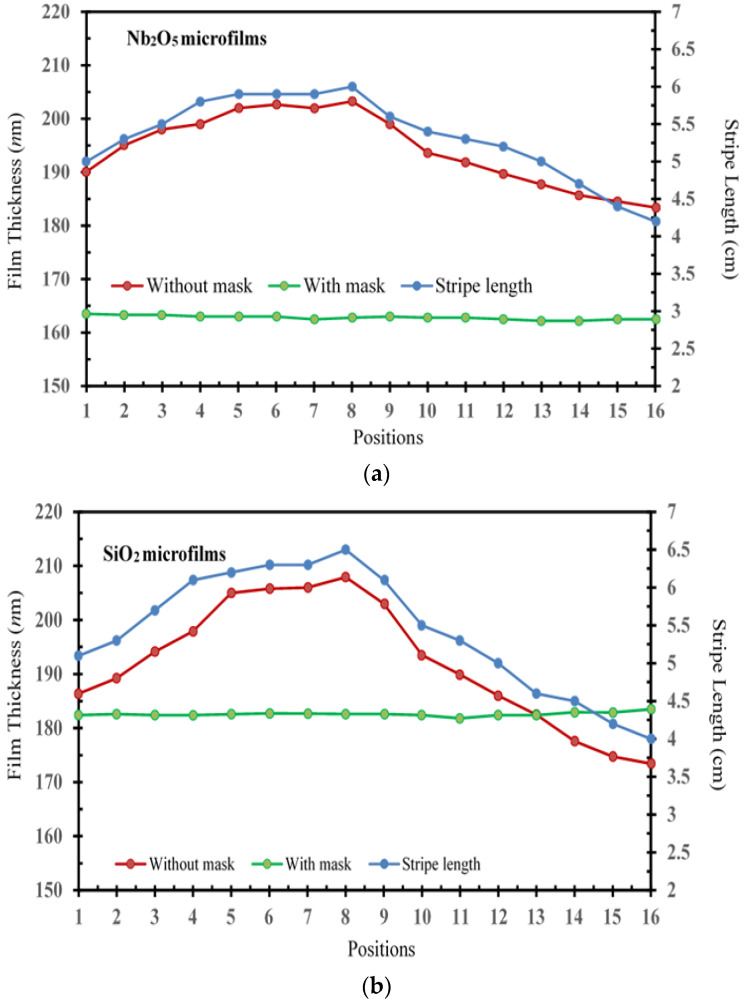
The adjustable stripe length of the shadow mask with the magnetron-IAD sputtering system and the corresponding film thicknesses as a function at the 16 positions for the (**a**) Nb_2_O_5_ and (**b**) SiO_2_ microfilm deposition.

**Figure 3 micromachines-15-00551-f003:**
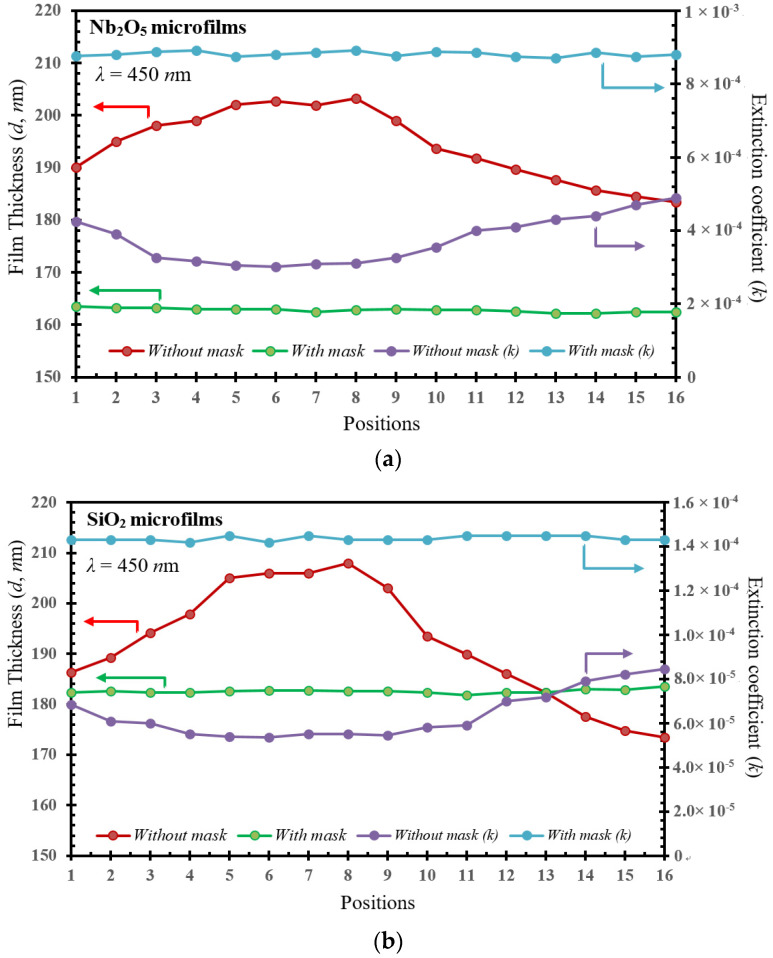
The comparison of the extinction coefficient (*k*) and the corresponding film thicknesses as a function at 16 different positions for (**a**) Nb_2_O_5_ and (**b**) SiO_2_ microfilms with/without shadow masks, respectively measured for the (*k*, *d*) parameter model.

**Figure 4 micromachines-15-00551-f004:**
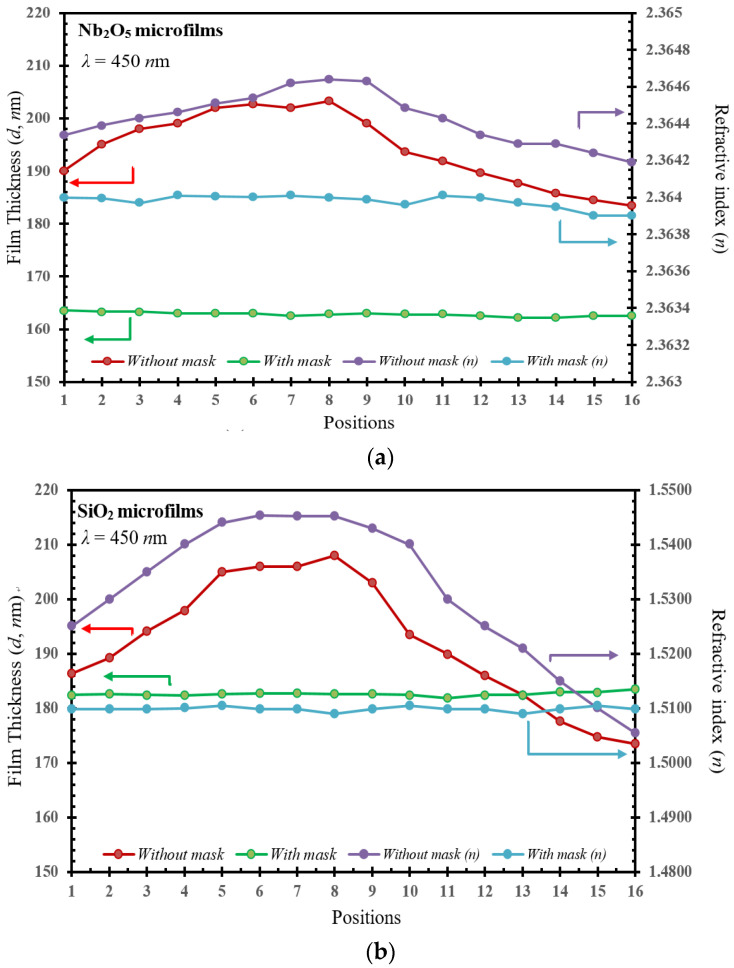
The comparison of the refractive index (*n*) and the corresponding film thicknesses as a function at 16 different positions for the (**a**) Nb_2_O_5_ and (**b**) SiO_2_ microfilm with and without shadow masks, respectively measured for the (*n*, *d*) parameter model.

**Figure 5 micromachines-15-00551-f005:**
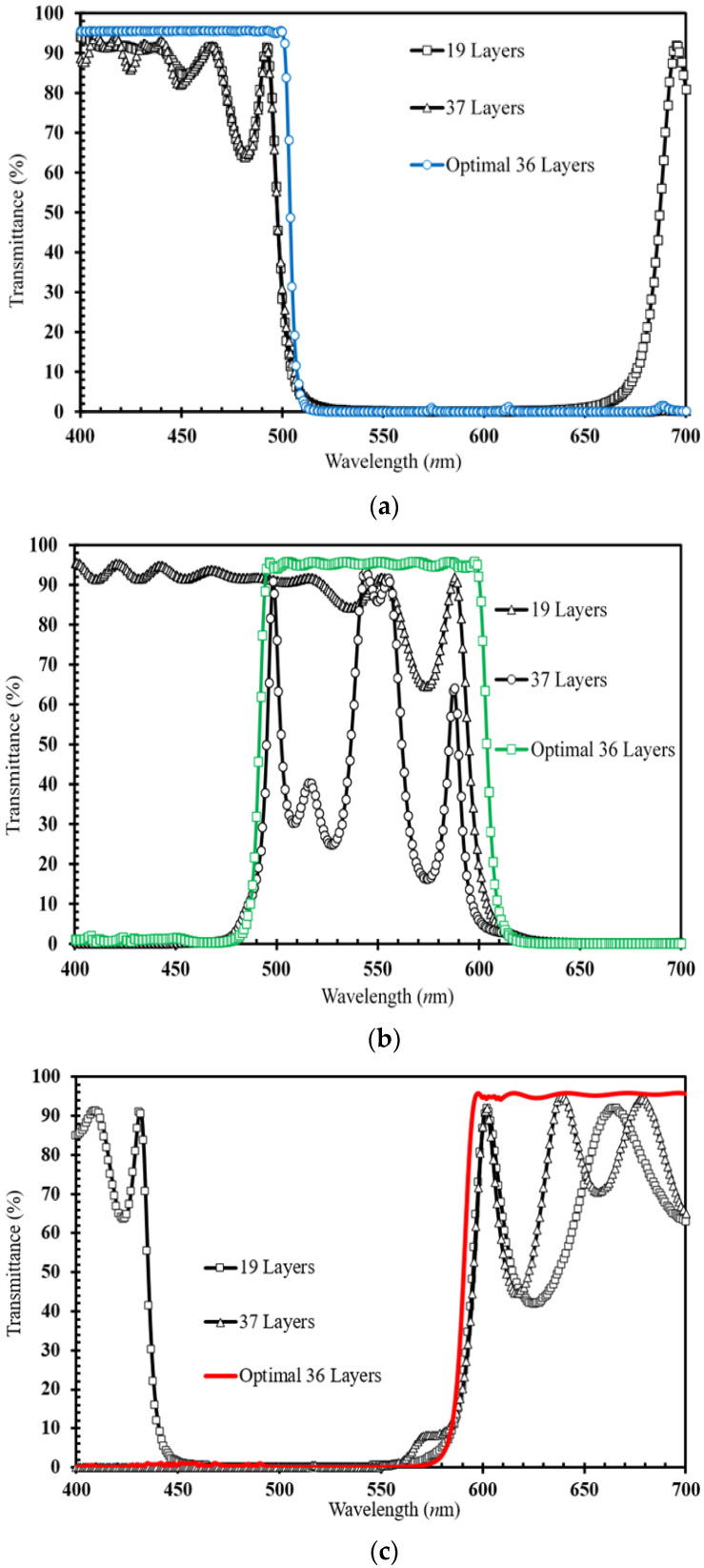
Transmittance comparisons between the experimental data and the calculated values from fitted (n, k, d) for CFs, respectively, at the (**a**) blue (400–500 nm wavelength), (**b**) green (500–600 nm wavelength), and (**c**) red (600–700 nm wavelength) narrow band-pass regions.

**Figure 6 micromachines-15-00551-f006:**
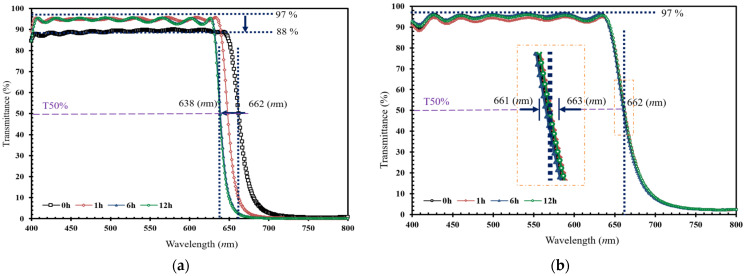
The comparison of the transmittance changes (in the visible range) of CFs fabricated by the magnetron-IAD sputtering system (**a**) without and (**b**) with a shadow mask, for environmental durability at 200 °C for 0, 1, 6, and 12 h. The cut-off wavelengths are quoted (with long-dash line) at the 50% (T = 50%) relative transmission.

**Table 1 micromachines-15-00551-t001:** The comparisons of Nb_2_O_5_ and SiO_2_ microfilms and the corresponding surface morphology (AFM images below) deposited with/without mask condition. Layer thicknesses of Nb_2_O_5_ and SiO_2_ microfilms are about 600 and 400 nm, respectively.

Nb_2_O_5_ Film (~400 nm thickness)					
Conditions	*R*max	*R*mean		*R*a	*R*q (RMS)
Without Mask	8.778 nm	3.044 nm		0.760 nm	0.974 nm
With Mask	8.710 nm	2.746 nm		0.603 nm	0.792 nm
SiO_2_ Film (~600 nm thickness)					
Conditions	*R*max	*R*mean		*R*a	*R*q (RMS)
Without Mask	5.36 nm	2.211 nm		0.488 nm	0.631 nm
With Mask	2.211 nm	1.139 nm		0.186 nm	0.242 nm
Nb_2_O_5_ Film (Without Mask)			Nb_2_O_5_ Film (With Mask)		
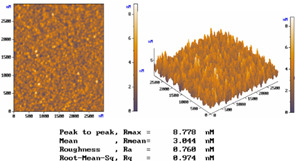			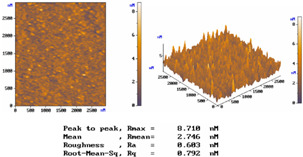		
SiO_2_ Film (Without Mask)			SiO_2_ Film (With Mask)		
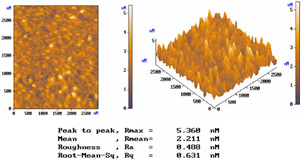			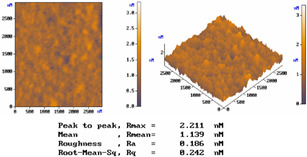		

**Table 2 micromachines-15-00551-t002:** The design parameters of CF application. (a) Each value of the optimal 36-layer thickness for the RGB-filter design, and (b) the schematic structure coating on a flexible HPC substrate (with 2.27–3.11 μm 36-layer thickness and 30 × 30 mm^2^ dimensions) by the developed IAD-sputtering shadow-mask system.

Layer No.	Materials	Blue (nm)	Materials	Green (nm)	Materials	Red (nm)
1	Nb_2_O_5_	88.68	Nb_2_O_5_	33.44	Nb_2_O_5_	51.37
2	SiO_2_	72.51	SiO_2_	208.32	SiO_2_	95.47
3	Nb_2_O_5_	77.72	Nb_2_O_5_	53.21	Nb_2_O_5_	46.69
4	SiO_2_	104.64	SiO_2_	88.38	SiO_2_	60.74
5	Nb_2_O_5_	50.16	Nb_2_O_5_	50.8	Nb_2_O_5_	58.74
6	SiO_2_	114.59	SiO_2_	31.57	SiO_2_	86.72
7	Nb_2_O_5_	63.81	Nb_2_O_5_	39.72	Nb_2_O_5_	53.51
8	SiO_2_	90.03	SiO_2_	87.3	SiO_2_	89.31
9	Nb_2_O_5_	58.52	Nb_2_O_5_	52.12	Nb_2_O_5_	51.35
10	SiO_2_	118.93	SiO_2_	87.14	SiO_2_	76.69
11	Nb_2_O_5_	50.42	Nb_2_O_5_	48.13	Nb_2_O_5_	85.81
12	SiO_2_	94.5	SiO_2_	29.07	SiO_2_	55.64
13	Nb_2_O_5_	68.67	Nb_2_O_5_	44.68	Nb_2_O_5_	55.37
14	SiO_2_	95.41	SiO_2_	88.81	SiO_2_	89.31
15	Nb_2_O_5_	59.86	Nb_2_O_5_	55.83	Nb_2_O_5_	47.77
16	SiO_2_	94.52	SiO_2_	70.86	SiO_2_	85.07
17	Nb_2_O_5_	68.84	Nb_2_O_5_	117.67	Nb_2_O_5_	117.88
18	SiO_2_	91.41	SiO_2_	99.89	SiO_2_	45.9
19	Nb_2_O_5_	61.59	Nb_2_O_5_	72.11	Nb_2_O_5_	47.98
20	SiO_2_	109.15	SiO_2_	97.2	SiO_2_	64.15
21	Nb_2_O_5_	76.77	Nb_2_O_5_	99.87	Nb_2_O_5_	52.39
22	SiO_2_	167.97	SiO_2_	109.35	SiO_2_	79.56
23	Nb_2_O_5_	75.99	Nb_2_O_5_	60.94	Nb_2_O_5_	20.62
24	SiO_2_	109.41	SiO_2_	129.06	SiO_2_	71.25
25	Nb_2_O_5_	64.96	Nb_2_O_5_	75.56	Nb_2_O_5_	47.44
26	SiO_2_	101.2	SiO_2_	141.68	SiO_2_	42.11
27	Nb_2_O_5_	86.55	Nb_2_O_5_	63.64	Nb_2_O_5_	46.29
28	SiO_2_	166.54	SiO_2_	98.73	SiO_2_	66.22
29	Nb_2_O_5_	68.15	Nb_2_O_5_	89.84	Nb_2_O_5_	54.71
30	SiO_2_	115.67	SiO_2_	130.8	SiO_2_	78.49
31	Nb_2_O_5_	64.19	Nb_2_O_5_	70.27	Nb_2_O_5_	63.17
32	SiO_2_	88.72	SiO_2_	105.79	SiO_2_	76.52
33	Nb_2_O_5_	76.02	Nb_2_O_5_	98.46	Nb_2_O_5_	51.69
34	SiO_2_	91.92	SiO_2_	32.49	SiO_2_	67.28
35	Nb_2_O_5_	69.49	Nb_2_O_5_	70.49	Nb_2_O_5_	42.73
36	SiO_2_	53.92	SiO_2_	30.19	SiO_2_	40.63
** Total (μm) **	** 3.11 **		** 2.86 **		** 2.21 **	
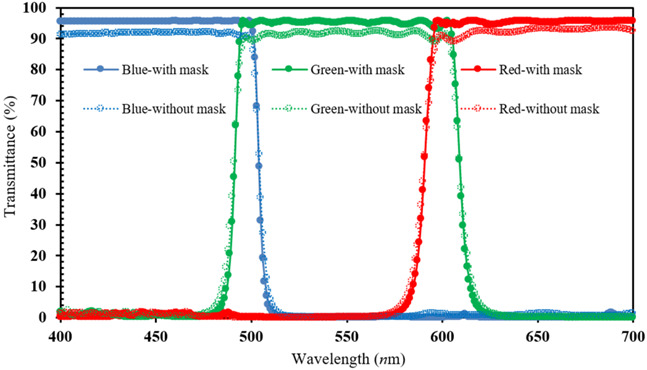						
(a)						
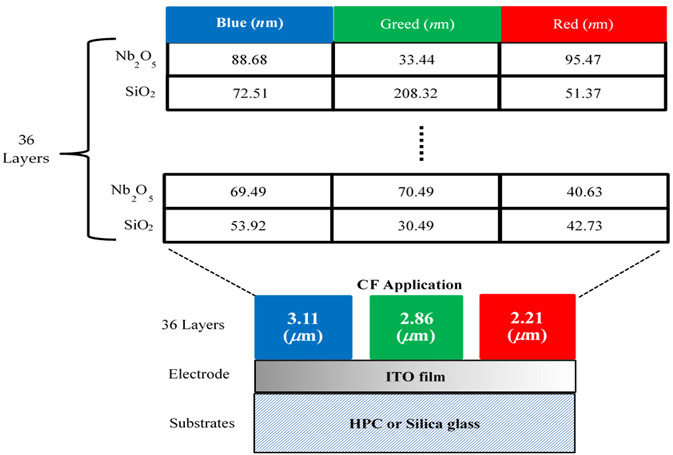						
(b)						

**Table 3 micromachines-15-00551-t003:** The RGB-CF transmittance comparison and the corresponding wavelength plot (below), with/without shadow-mask, by the magnetron-IAD sputter.

	Filters	Blue Filter (400~500 nm)	Green Filter (500~600 nm)	Red Filter (600~700 nm)
Condition				
Thickness		3.11 μm	2.86 μm	2.21 μm
With Mask		95.5041%	95.6174%	95.7046%
Without Mask		91.4657%	92.6601%	93.5871%

## Data Availability

The data presented in this study are available on request from the corresponding author. The data are not publicly available due to privacy and ethical restrictions.

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
