# Peer review of "Enhancing Thickness Uniformity of Nb2O5/SiO2 Multilayers Using Shadow Masks for Flexible Color-Filtering Applications"

_micromachines, 2024, doi:10.3390/mi15040551_

Round 1
Reviewer 1 Report
Comments and Suggestions for Authors
micromachines-2940949
This is an interesting work described the effects of one mask in front of the ion source on the morphology and optical properties of NbO/SiO multilayers. Before acceptation, some modifications and more details should be made.
(1) Please provide more details in Experiments and Measurements part. (i) the
rotation speed of the substrates; (ii) the size of substrates; (iii) the materials of the shadow mask, stainless steel? or PC? (iv) the mask is grounded or floating? (v) the discharge voltage of the ion source. (vi)
(2) Line 110, Page 3. Which deposition process has a 13 nm/min rate? According to Figure 2, the film thickness has a noticeable decrease, especially for Nb2O5, from 205 nm to 165 nm (~20%) for example. So please give the deposition time of the data in figure2.
(3) Please provide the refractive index (n&k) in the range of 400-700 nm wavelength of the SiO2 and Nb2O5 with and without mask.
(4) The data (or the structure) used in Figure 4 is different from that of both in Figure 3, that are 36-layer for green, blue, and red CFs, why? What is the structure (or how many layers and the thickness of each layer) of the curves in Figure4.
(5) Stability is closely related to its growth structure for sputtered films. Please provide cross-sectional SEM images of the film used for durability test.
(6) Please the authors give an explanation why the shadow mask has such a noticeable effect on the film growth. Plasma distribution? Energy and flux of O ions? Or others?
Author Response
MS #: (Micromachines-2940949)-R1
Title: Enhancing Thickness Uniformity of Nb2O5/SiO2 Multilayers using Shadow Masks for Flexible Color-Filtering Applications.
Dr. Eric Jiao (Ms. Faye Zou) with all reviewers,
Guest Editor (Section Managing Editor), Micromachines,
(IF 3.4, ISSN 2072-666X)-Special Issue ",
Micromachines Editorial Office, St. Alban-Anlage 66, 4052 Basel, Switzerland
MDPI Wuhan Micromachines Editorial Office,
E-Mail: micromachines@mdpi.com. Tel. +41 61 683 77 34; Fax: +41 61 302 89 18
Floor 54, Wuhan Hnag Lung Plaza Office Tower, 430030 Wuhan, China.
Tel.: +86 27 8780 8658;
Email: faye.zou@mdpi.com (Ms. Zou), eric.jiao@mdpi.com (Eric Jiao, assistant); Micromachines Editorial Office <micromachines@mdpi.com>, chenhc@mail.nsysu.edu.tw.
website: https://www.mdpi.com/journal/micromachines/instructions; Submit to the issue at: https://www.mdpi.com/journal/micromachines/special_issues/AQN7IXOV1U.
Dear All reviewers/Editor & Ms. Zou,
Thanks for your e-mail dated at April 03, 2024.
Enclosed please find manuscript files of the paper entitled above, by T.-Z. Li et.al. In the revised one (R1micromachines-2940949Revised.doc), all the comments of the reviewers have been overcome and marked with bold-red words. Hope this revised one can be accepted to publish on the micromachines journal (MDPI).
In addition, the special revisions per reviewers’ comments (AnswerQueries(micromachines2940949 R1).pdf) has been attached for editors’ and reviewers’ convenience to check Your kind assistance in dealing with this matter is my most appreciated.
Best regards from Sincerely Yours,
Jyh-Jier HO, Ph.D., 04/10/2024

Reviewer 2 Report
Comments and Suggestions for Authors
This paper shows some interesting results of the improvement to uniformity and surface quality of optical films using a variable slit width mask. However, there is additional information and corrections needed and also some clarification of the English.
List of comments (not in order of importance)
Abstract
Line 39 “by substitution of surface adhesive properties” The meaning of this phrase is not clear.
Line 40 “flexible” Do you mean application on flexible plastic substrates?
Experiments and measurement
You have to explain certain what the effect of the oxygen ion bombardment on the films is. Are the Nb and Si oxide films stoichiometric without ion bombardment? Presumably the effect of ion bombardment is to etch away the oxide film and therefore the slit widths are adjusted so that the final film thickness is uniform? How were the slit widths calculated? Do they relate mathematically to the distance of the ion source from the substrate? Were they found by trial and error? How thick is the shadow mask?
Does ion bombardment and etching have any effect on the film properties (apart from roughness and thickness) such as stoichiometry? This needs some data measurement?
How do you control the magnetron target poisoning and film stoichiometry?
What is the target current for each target? Are the cylindrical targets rotating? Is the target voltage DC or pulsed DC?
Line 68 and others Should stripe length not be stripe width?
Lines 99-102 This sentence is not clear.
Line101 “dissolve silica glass” Explain what this is.
Lines 107-108 How does oxygen gas affect substrate temperature? How is substrate temperature controlled?
Line 164 The doi reference for this paper does not work
Line 168 How can the Nb2O5 thickness variations range from 19.86 to 1.33 nm? Surely you can only quote one value for this? Similarly for SiO2.
Figure 2 Indicate which curves relate to which Y-axis.
Line 193 “as shown in the bottom-left axis of Figure 2.” Figure 2 has no data on surface roughness.
Lines 200-201 “it can be found that the thickness of the microfilm first increases corresponding to the
position number from 1 to 8; and then decreases from the 9th to the 16th position.” Comment on the reason for this.
Line 202 “measuring the average-thickness variations” Should this be roughness rather than thickness variations?
Line 222 “the layering process causes particles to settle between and on top of each layer.” Explain whyu this occurs.
Lines 355-359 It is not enough to say there was little change. How much change was there? Measurement data is needed here.
Comments on the Quality of English LanguageSome of the language is unclear (presumably because of the difficulty of writing in English and there is some clumsy constructions. The paper needs proof read to correct these either by the editors or by the authors.
Author Response
MS #: (Micromachines-2940949)-R1
Title: Enhancing Thickness Uniformity of Nb2O5/SiO2 Multilayers using Shadow Masks for Flexible Color-Filtering Applications.
Dr. Eric Jiao (Ms. Faye Zou) with all reviewers,
Guest Editor (Section Managing Editor), Micromachines,
(IF 3.4, ISSN 2072-666X)-Special Issue ",
Micromachines Editorial Office, St. Alban-Anlage 66, 4052 Basel, Switzerland
MDPI Wuhan Micromachines Editorial Office,
E-Mail: micromachines@mdpi.com. Tel. +41 61 683 77 34; Fax: +41 61 302 89 18
Floor 54, Wuhan Hnag Lung Plaza Office Tower, 430030 Wuhan, China.
Tel.: +86 27 8780 8658;
Email: faye.zou@mdpi.com (Ms. Zou), eric.jiao@mdpi.com (Eric Jiao, assistant); Micromachines Editorial Office <micromachines@mdpi.com>, chenhc@mail.nsysu.edu.tw.
website: https://www.mdpi.com/journal/micromachines/instructions; Submit to the issue at: https://www.mdpi.com/journal/micromachines/special_issues/AQN7IXOV1U.
Dear All reviewers/Editor & Ms. Zou,
Thanks for your e-mail dated at April 03, 2024.
Enclosed please find manuscript files of the paper entitled above, by T.-Z. Li et.al. In the revised one (R1micromachines-2940949Revised.doc), all the comments of the reviewers have been overcome and marked with bold-red words. Hope this revised one can be accepted to publish on the micromachines journal (MDPI).
In addition, the special revisions per reviewers’ comments (AnswerQueries(micromachines2940949 R1).pdf) has been attached for editors’ and reviewers’ convenience to check Your kind assistance in dealing with this matter is my most appreciated.
Best regards from Sincerely Yours,
Jyh-Jier HO, Ph.D.

Reviewer 3 Report
Comments and Suggestions for Authors
The authors have presented an improved Nb2O5/SiO2 thickness uniformity by using shadow masks. The manuscript is well-written and the results were well-organized. The authors mentioned the thickness uniformity was improved for both Nb2O5/SiO2 films with masks. However, the impact of shadow masks on SiO2 was much bigger than on Nb2O5. Can the authors explain the reason why? The authors did mentioned the surface roughness reduced more on SiO2 than on Nb2O5, but there was still no explanation on why this difference was observed.
The manuscript can be accepted after the comment is address.
Comments on the Quality of English Language-
Author Response

(The authors gave the same response as above.)

Round 2
Reviewer 1 Report
Comments and Suggestions for Authors
Authors have substantially modified their manuscript, and I suggest it can be accepted in the present form.
Author Response
April 17, 2024
MS #: (Micromachines-2940949)-R2
Title: Enhancing Thickness Uniformity of Nb2O5/SiO2 Multilayers using Shadow Masks for Flexible Color-Filtering Applications.
Dr. Eric Jiao (Mr. Michael Han and Ms. Faye Zou) with all reviewers,
Guest Editor (Section Managing Editor), Micromachines,
(IF 3.4, ISSN 2072-666X)-Special Issue ",
Mr. Michael Han (Assistant Editor)
Micromachines Editorial Office, St. Alban-Anlage 66, 4052 Basel, Switzerland
MDPI Wuhan Micromachines Editorial Office,
E-Mail: micromachines@mdpi.com. Tel. +41 61 683 77 34; Fax: +41 61 302 89 18
Floor 54, Wuhan Hnag Lung Plaza Office Tower, 430030 Wuhan, China.
Tel.: +86 27 8780 8658;
Email: michael.han@mdpi.com (Mr. Han), faye.zou@mdpi.com (Ms. Zou), eric.jiao@mdpi.com (Eric Jiao, assistant); Micromachines Editorial Office <micromachines@mdpi.com>, chenhc@mail.nsysu.edu.tw.
website: https://www.mdpi.com/journal/micromachines/instructions; Submit to the issue at: https://www.mdpi.com/journal/micromachines/special_issues/AQN7IXOV1U.
Dear All reviewers/Editor & Mr. Han,
Thanks for your e-mails (revision request) dated at April 15 and 17, 2024.
Enclosed please find manuscript files of the paper entitled above, by T.-Z. Li et.al. In the revised one (R2CFiltersMicromachinesTemplate(2940949).docx), all the comments of the reviewers have been overcome and marked with bold-red words. Hope this revised one can be accepted to publish on the micromachines journal (MDPI).
In addition, the special revisions per reviewers’ comments (AnswerQueries(micromachines2940949 R2).pdf) has been attached for editors’ and reviewers’ convenience to check Your kind assistance in dealing with this matter is my most appreciated.
Best regards from Sincerely Yours,
Jyh-Jier HO, Ph.D.

Reviewer 2 Report
Comments and Suggestions for Authors
I thank the authors for their clarifications and additions. However, I still have some concerns.
Firstly, I apologise for my comments recommending the use of “stripe width” rather than “stripe length for the mask. The authors are correct in that “stripe length” is the correct term. However, for clarity, the authors should include a diagram showing the orientation of the mask relative to the targets to show which areas undergo more bombardment.
I note that the authors have added important information about the effect of ion bombardment on refractive index. The effect of oxygen ion bombardment on the films is critically important. Since the paper describes the equalisation of the thickness across the substrate, it is clear that the greater total ion bombardment due to the longer stripe causes greater erosion of the film. The fact also that the roughness is improved is also a feature which implies that the ion bombardment has a beneficial effect. I appreciate that the authors have not investigated these aspects specifically but they should comment on prior published work showing these effects.
Specific points
Line 11 and elsewhere “stainless” should be “stainless steel”
Line 41 “hardness polycarbonate” Should this be “hard polycarbonate” or is “hardness polycarbonate” a recognised term?
Line 168 “It can contribute that shadow-mask sizes are typically adjusted for the uniform optimization via the blocking of sputtered materials [17].” This is clearly the case for masks placed between the target and the substrate which limit the amount of material deposited but in this case the mask is between the source and the substrate ion to control the amount of material removed by the ion bombardment so the reference is not relevant here.
Fig 2(b) I note that the SiO2 film thickness with the mask is higher in positions 13-16 than without the mask. This needs explanation.
Author Response

(The authors gave the same response as above.)
